# Prediction model and technical and tactical decision analysis of women's badminton singles based on machine learning

**Hanguang Yuan[1], Yaodong Wang[1]\*, Kairan Yang[2], Yulu Bin[3]**

**1** Department of Physical Education, University of Mining and Technology (Beijing), Beijing, China, **2** College of Physical Education, Hunan Normal University, Hunan, China, **3** College of Art, Beijing Sport University, Beijing, China

\* 22502813@qq.com

**Data Availability Statement:** All relevant data are within the manuscript and its Supporting Information files.Video data from open badminton

## Abstract

In the Paris Olympic cycle, South Korean women's athlete An Se-young rose to the top of the 2023 BWF Olympic points with a win rate of 89.5%. With An Se-young as the subject, this paper aims to carry out technical and tactical analysis of women's badminton singles and formulate a prediction model based on machine learning. Firstly, An's technical and tactical statistics are analyzed and presented in a proposed "three-stage" data classification method. Secondly, we improve our "three-stage" machine learning dataset using video analysis of 10 matches (21 point games) where An Se-young faced off against four other players ranked in the top five of the World Badminton Federation (BWF) in week 44 of 2023. Finally, we establish a prediction model for the scoring and losing of points in the women's badminton singles based on the 'Decision tree', 'Random forest', 'XGBoost', 'Support vector' and 'K-proximity' algorithms, and analyze the effectiveness of this model. The results show that the improved data classification is reasonable and can be used to predict the final score of a match. When the support vector machine uses the RBF function kernel, the accuracy reaches its highest at 87.5%, and the consistency of this prediction model is strong. An's playstyle is sustained and unified; she does not seek continuous pressure, but rather exploits and maximizes her aggression following any mistake made by her opponents, immediately utilizing assault methods such as kills or dives, often resulting in the conversion of points during the subsequent 2–3 strikes.

## 1 Introduction

Women's singles in Badminton has always been in the spotlight, and in recent years it has produced many strong players, showing the current situation of "multi-power competition". However, the recent rise of South Korea's An Se-young has brought new changes to this competitive landscape. In the 2023 Paris Olympics points year, An Se-young performed well on the Badminton World Federation (BWF) Olympic standings, increasing her win rate to 89.5% with 77 wins and 9 losses (The data comes from http://www.youtube.com). This win

competition official website, https://bwfbadminton.com/.

**Funding:** The author(s) received no specific funding for this work.

**Competing interests:** The authors have declared that no competing interests exist.

percentage reflects Se-young's consistency and strong competitiveness in the game, making her a key opponent to watch in the women's singles event at the Paris Olympics.

There are few studies on An, but relevant studies have been carried out in the field of badminton. In recent years, with An as the research object, numerous studies have been conducted to analyze the badminton game in detail, which has revealed the intricate technical and tactical styles of the world's top badminton players, as well as the characteristics of their serve performance in competition. Among others, Gómez-Ruano (2020) explored the serving patterns of women's badminton medalists at the 2016 Rio Olympics, taking into account both the record and time variables, to depict and identify the serving characteristics of medalists in elite women's badminton events [1]. Meanwhile, Torres-Luque, G (2020) devoted their studies to the analysis of statistical differences between men's and women's badminton singles matches in the London 2012 and Rio 2016 Olympic Games [1]. Gomez (2020) identified contextual variables associated with the occurrence of long rounds while examining time-related and technical parameters during games [2]. He further considered factors such as the context of the game and the gender of the players to delve deeper into the performance differences between long rounds and then their subsequent rounds. In addition, Xu (2023) focused on the women's singles badminton event at the 2020 Tokyo Olympics, analyzing in-depth the performance of the competition phases and medalists to clearly define the technical and tactical characteristics of the world's top badminton players [2]. However, current research focuses mainly on qualitative analysis of their hitting characteristics through statistical methods or evaluating them only for a single technique. This type of analysis relies to some extent on experience [3], and although it can characterize playing styles, it has limitations in quantifying tactical combinations as well as predicting ball changes. The latter, on the other hand, plays a crucial role in developing targeted tactics for opponents. Therefore, conducting research on badminton tactical prediction models and decision-making analysis holds significant practical importance.

Common methods for tactical combination analysis and shot prediction in the field of badminton primarily include probabilistic statistical analysis and the construction of predictive models based on machine learning. In the field of mathematical statistics, probabilistic statistical models and machine learning models are commonly used for regularity analysis and prediction. Probabilistic statistical models are usually used when things have a strong theoretical basis or linear law scenario [4], However, in the arena, techniques and tactics are difficult to arrange and combine through simple theories, and clinical play is characterized by a variety of techniques. As a result, it is generally difficult to make effective predictions using probabilistic statistical models, and tactical combinations are often analyzed in a qualitative phase [5]. Machine learning, as a branch of artificial intelligence, utilizes different algorithms to extract patterns through regular autonomous learning of data in order to predict future events. Machine learning is widely used in natural language processing, data mining, finance, medicine and other fields [6]. In the field of sports, Kaustubh Milind Kulkarni (2021) et al analyzed the batting performance of men's table tennis using two-dimensional human gesture recognition to reveal the variations and effectiveness of athletes' technical and tactical use in different game sequences. The results indicate that neural network models can contribute to the development of computer vision-based sports analysis systems [7]. Olivier Dieu (2020) used the random forest algorithm to analyze the video of multiple badminton players' matches to help coaches subjectively judge the reasonableness of athletes' technical choices with objective data. The results demonstrate that utilizing player tracking data and machine learning algorithms can identify and classify actions in small ball sports, with the random forest algorithm exhibiting the best performance [8]. Tindaro Bongiovanni (2021) and others, on the other hand, focused on the influence of physiological indicators on soccer players' athletic performance and constructed the Extra Tree Return model to predict the aerobic capacity of adolescent

athletes using their physiological indicators as baseline data. The results show that the prediction accuracy of the random forest algorithm surpasses that of subjective probability [8].

An Se-young's performance in 2023 disrupted the long-standing "multi-power rivalry" in women's singles badminton. Her "phenomenal" annual win rate of 89.5% not only highlights the superiority of her tactical combinations but also demonstrates her competitive stability. Therefore, analyzing the shot patterns and tactical combinations of top female badminton players, with An Se-young as the subject, holds significant research value. Machine learning, through various algorithmic models, enables systematic autonomous learning from data, making it an effective means for quantitatively analyzing players' playing habits, tactical combinations, and shot predictions. The selection of an appropriate database and suitable machine learning models is crucial for prediction accuracy, necessitating in-depth research.

To summarize, this paper intends to carry out the research of machine learning-based prediction models in women's badminton singles, while also analyzing An Se-young technical and tactical decisions. Firstly, An's technical and tactical statistics are analyzed and presented in a proposed "three-stage" data classification method. Secondly, we improve our "three-stage" machine learning dataset using video analysis of 10 matches (21 point games) where An Se-young faced off against four other players ranked in the top five of the World Badminton Federation (BWF) in week 44 of 2023. Finally, we establish a prediction model for the scoring and losing of points in the women's badminton singles based on the 'Decision tree', 'Random forest', 'XGBoost', 'Support vector' and 'K-proximity' algorithms, and analyze the effectiveness of this model. The above study intends to provide an example quantitative analysis of badminton women's singles skills and tactics.

## 2 Research methodology

### 2.1 Decision trees

Decision trees are intuitive machine learning algorithms for learning simple decision rules from data to infer target values. They perform feature comparisons of internal nodes and move along the branches based on these comparisons until they reach the leaf nodes, which give predictions [8]. The tree building process is as follows.

**(1) Information gain.**   Information gain is a criterion used in decision tree algorithms to select optimal segmentation features [9]. It determines the validity of a feature by calculating the entropy reduction of the data set under that feature, where entropy is a measure of uncertainty in the data set. A higher information gain means that the feature is more important in the classification problem. Specific formula for information gain:

$$Gain(D, a) = Entropy(D) - \sum_{v \in Values(a)} \frac{|D_v|}{|D|} Entropy(D_v) \tag{1}$$

Where D is the dataset, $a$ is the feature, and $Values(a)$ is all possible values of feature $a$.

**(2) Gini impurity.**   Gini impurity is another method used to select features in decision tree algorithms. It measures the probability that two randomly selected samples in a data set belong to different categories, the lower the Gini impurity, the higher the purity of the data, i.e., the better the classification.

$$Gini(D) = 1 - \sum_{i=1}^{m} p_i^2 \tag{2}$$

where $p_i$ is the relative frequency of class $i$ in dataset D.

**(3) Pruning.**   Pruning is a technique used in decision tree algorithms to avoid overfitting. It regularizes the model by reducing the complexity of the tree (e.g., by reducing the depth of

the tree or the number of nodes), thus improving the model's ability to generalize to unknown data. Pruning can balance the complexity and performance of a model by setting a regularization parameter.

$$C_\alpha(T) = \sum_{t=1}^{|T|} N_t H_t(T) + \alpha |T| \tag{3}$$

where $C_\alpha(T)$ is the cost of the tree after pruning, $N_t$ is the number of samples in node t, $H_t(T)$ is the impurity of node t, and $|T|$ is the total number of nodes in the tree, but $\alpha$ is the regularization parameter.

The prediction process is as simple as traversing the tree up to the leaf node and then returning the value of the leaf node. Integration methods such as Random Forest and Gradient Boosting construct multiple such trees to improve accuracy.

## 2.2 Random forests

Random Forest is an integrated machine learning algorithm that improves the accuracy and robustness of a model by constructing multiple decision trees and synthesizing their predictions. Each tree is trained independently, using a randomly selected subset of features and samples [10]. The key steps of a random forest can be expressed by the following equation:

**(1) Self-sampling (random selection of samples).**

$$D_i \sim Bootstrap(D) \tag{4}$$

where D is the original dataset and $D_i$ is the training dataset for the $i$ tree.

**(2) Feature subset selection (random selection of features).**

$$F_i \subset F \tag{5}$$

where $F$ is the set of all features and $F_i$ is a randomly selected subset of the features in F.

**(3) Integration of decision trees.**

$$f(x) = \frac{1}{N} \sum_{i=1}^{N} T_i(x, D_i, F_i) \tag{6}$$

For the regression problem, $T_i(x, D_i, F_i)$ is the prediction of the $i$ tree for the input $x$. N is the total number of trees.

**(4) Majority vote.**   Majority voting is a decision-making method used in integrated learning for classification problems that aggregates the predictions from multiple models (e.g., decision trees) and selects the category with the highest number of occurrences as the final classification result. This approach enhances model stability and accuracy because it combines the predictions of multiple models, thereby reducing the impact of chance errors in a single model.

$$f(x) = mode\{T_i(x, D_i, F_i)\} \tag{7}$$

For the classification problem, the output is the most common class of all tree predictions.

The efficiency and generalization capabilities of random forests make them a powerful tool for solving a variety of classification and regression problems. Although the model is larger and less explanatory than a single decision tree, it performs well in many practical applications.

## 2.3 XGBoost

XGBoost is an integrated learning algorithm that builds a more stable or stronger predictive model by combining multiple weak predictive models (usually decision trees) [11]. Each newly added model tries to correct the errors of the previous model, which is achieved by means of gradient boosting [12]. The XGBoost algorithm tries to add a new tree at each step that minimizes the loss function of the overall model after adding this tree.

The objective function of XGBoost consists of two parts: a training loss and a penalty term for model complexity. The training loss measures how well the model fits the training data, while the complexity penalty term controls the complexity of the model to prevent overfitting. The objective function can be expressed as function (8), where $L(\Theta)$ is the training loss function, $\Omega(\Theta)$ is the penalty term for model complexity, and $\Theta$ denotes the parameters of the model.

$$Obj(\Theta) = L(\Theta) + \Omega(\Theta) \tag{8}$$

XGBoost supports many types of loss functions that can be used for regression (e.g., squared loss), classification (e.g., log loss), and many more machine learning tasks. For example, in binary classification problems, the loss function is usually a log-loss, where y is the true label and $\hat{y}$ is the predicted probability.

$$L(y, \hat{y}) = -y \log \log(\hat{y}) - (1 - y)\log \log(1 - \hat{y}) \tag{9}$$

The model complexity $\Omega(\Theta)$ is calculated based on the depth of the tree and the number of leaf nodes and is used to penalize overly complex models, where T is the number of trees, $w$ is the leaf node weight of the tree, and $\gamma$ and $\lambda$ are regularization parameters.

$$\Omega(\Theta) = \gamma T + \frac{1}{2}\lambda \parallel w \parallel^2 \tag{10}$$

XGBoost uses a series of optimizations, including approximation algorithms, sparse-aware algorithms, and a unique tree-learning algorithm, to improve the efficiency and accuracy of the algorithm. It also supports a variety of parallel and distributed computations, making training on large datasets faster.

## 2.4 Support vector machines

A support vector machine (SVM) is an algorithm for classification and regression that works by finding the best separating hyperplane between data points [13].

**(1) Decision functions.**

$$f(x) = sign(w \cdot x + b) \tag{11}$$

where $w$ is the normal vector to the hyperplane, $x$ is the eigenvector, $b$ is the bias term, and the sign function returns the sign of the input.

**(2) Maximizing edges.**  Maximizing edges means finding the best classification hyperplane while maintaining the maximum distance between the classification hyperplane and all data points. This is done to improve the generalization ability of the model, i.e., to achieve better classification results even on unseen data.

$$\max_{w,b} \frac{1}{|w|} \tag{12}$$

Subject to the following constraints

$$y_i(w \cdot x_i + b) \geq 1, \forall i \tag{13}$$

where $y_i$ is the label of sample i and $x_i$ is the feature vector of sample i.

**(3) Kernel function.** The role of the kernel function in a support vector machine is to enable the original feature space to be mapped to a higher dimensional space by a non-linear transformation, so that it is easier to perform an efficient linear segmentation with hyperplanes in this new feature space. This allows SVMs to deal with complex datasets that are linearly indistinguishable in the original space, thus improving the applicability of the algorithm and classification accuracy. In short, the kernel function enables SVMs to handle data with nonlinear relationships.

a. linear kernel: $K(x_i, x_j) = x_i \cdot x_j$

b. polynomial kernel: $K(x_i, x_j) = (\gamma x_i \cdot x_j + r)^d$

c. RBF kernel: $K(x_i, x_j) = exp\,exp(-\gamma|x_i - x_j|^2)$

d. Sigmoid nucleus: $K(x_i, x_j) = tanh\,tanh(\gamma x_i \cdot x_j + r)$ where, $\gamma$, $d$, and $r$ are arguments to the kernel function.

**(4) Lagrange multiplier.** The role of Lagrange multipliers in support vector machines is for solving optimization problems, especially when dealing with optimization problems with constraints. It simplifies the computational process of finding the optimal hyperplane by introducing additional variables (Lagrange multipliers) that transform the original constrained optimization problem into an unconstrained optimization problem. In SVM, this method is used to simultaneously maximize edges and satisfy classification constraints to determine support vectors and optimal segmentation hyperplanes.

$$L(w, b, \alpha) = \frac{1}{2}|w|^2 - \sum_{i=1}^{N} \alpha_i[y_i(w \cdot x_i + b) - 1] \tag{14}$$

Minimize $L(w,b,a)$, subject to $\alpha_i \geq 0$ and take the derivative for each $\alpha_i$ and set it to zero.

## 2.5 K-nearest neighbor algorithm

K Nearest Neighbor Algorithm (KNN) is a simple algorithm that performs classification or regression by finding the nearest neighbors of a data point in the feature space [14]. In KNN, there is no step of learning the model parameters; instead, it uses the entire training dataset to make predictions.

For classification, the prediction formula for KNN is:

$$y_{pred}(x) = mode\{y_i | x_i \in K - nearest\ neighbors\ of\ x\} \tag{15}$$

For regression, the prediction is:

$$y_{pred}(x) = \frac{1}{K}\sum_{x_i \in K-nearest\ neighbors\ of\ x} y_i \tag{16}$$

Where $y_i$ is the label of the training sample and $K$ is the number of neighbors. Distances are usually calculated using the Euclidean distance:

$$d(x, x_i) = \sqrt{\sum_{j=1}^{n} (x_j - x_{ij})^2} \tag{17}$$

Where $n$ is the number of features, $x_j$ is the value of a feature and $x_{ij}$ is the value of that feature in the training sample. Choosing the right K-value and distance metric is critical to the

performance of the algorithm. KNN is suitable for problems with small datasets and unknown data distribution, but may be limited by computational and storage efficiency when dealing with large-scale datasets.

## 3 Study objects and dataset construction

### 3.1 Object of study

As previously mentioned, An Se-young topped the 2023 BWF Olympic Points Table with an impressive win rate of 89.5%. Therefore, in our research on the prediction model for women's singles badminton, we aimed to analyze data on an annual basis, focusing on a single variable, with An Se-young as the subject. We compiled data on all her matches in 2023 against other players ranked 2–5 in the world, totaling 18 matches. To avoid interference from other factors, we excluded matches where An Se-young or her opponents participated while injured, as well as tournaments below the 300 level. After balancing the participation frequency of other top 2–5 ranked players, we ultimately selected 10 matches as the dataset for this study. Admittedly, the model constructed in this study has certain limitations, such as a small sample size, simplistic criteria for personal style classification, and the lack of consideration for differences in players' handedness. In future research, we can address these limitations by expanding the sample dataset, improving the accuracy of personal style classification, and conducting more in-depth statistical analysis with large data samples to provide more accurate and targeted pre-match tactical advice.

Ten representative matches were selected as the source of data for this study. In terms of An Se-young's opponents, players ranked 2–5 in the WBF Olympic Points Ranking (as of 2023) were selected, including Chen Yu Fei, Xie Yamaguchi, Tzu-Ying Dai, and Carolina Marin. The matches between An Se-young and her opponents were all of a very high competitive level. Except for An Se-young, the other four athletes had mutual victories and defeats in the current year, so it is beneficial to reduce the modeling bias caused by technical differences. From the 10 matches, a total time of 10h32min was recorded, with the actual game time recorded as 140min49s. This includes finals and/or semifinals from the Indonesia Open, China Open, World Championships, Asian Championships and Asian Games organized by the Badminton World Federation (BWF) in 2023, in which An Se-young participated.

### 3.2 Dataset construction

**3.2.1 Data source.** The badminton competition videos used in this study were obtained from the YouTube video sharing platform (http://www.youtube.com), which is a video resource that can be used for public scientific research (for non-profit purposes).

**3.2.2 Data selection.** An Se-young has a strong ability to use shot placement and tactics to score through multiple deliberately placed shots. She usually combines a 2–3 shot sequence to convert certain playing tactics into points. This means it is difficult to reflect the characteristics of An's playing style and the way she changes the ball path in the multi-shot phase if the statistics are conducted using the traditional serve-receive segment, restriction segment and multi-shot segment as the statistical dataset. Based on this, the following improvements were made in this study:

1. On the basis of the three-stage, the data on serving errors were further screened, more than 2 shots of valid data were selected for collection, and the area where the scored and lost balls landed was added (e.g., if the ball went out of bounds or didn't go over the net, it was judged to be a forced or unforced error);

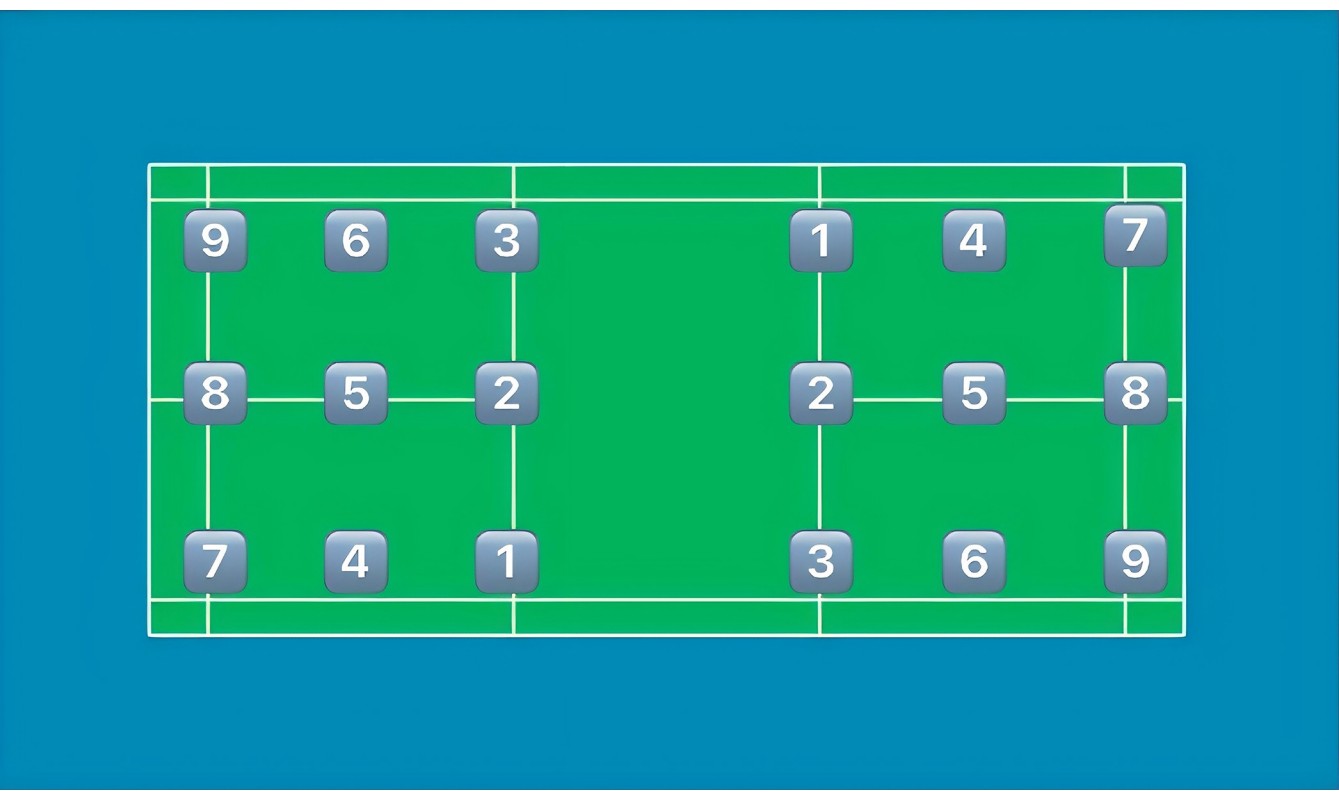

**Fig 1. The serving position and its corresponding number.**

2. In view of the characteristics of An Se-young's playing style, the detailed statistics of the playing tactics of the last 3 shots, including the location of the ball and the techniques used, a visualization platform proposed by Wei-Yao Wang (2023) has been deployed to illustrate the variability of analysis cases from Shuttle Set for coaches to delve into players' tactical preferences with human-interactive interfaces, which was also used by national badminton teams during multiple international high-ranking matches [15], are shown in Fig 1 and Table 1. The location of the ball and its corresponding number are shown in Fig 1 and Table 1, and the names of the techniques used and their corresponding numbers are shown in Table 2.

**3.2.3 Statistical methods.** The data is gained through game replays from BWF Fansite (bwfbadminton.com). Statistics such as game times, the total number of strikes, An Se-young's

**Table 1. The serving position and its corresponding number.**

| No. | The Serving Position | No. | The Serving Position |
|---|---|---|---|
| 1 | Forehand net | 6 | Backhand baseline |
| 2 | Center net | 7 | Forehand backcourt |
| 3 | Backhand net | 8 | Center backcourt |
| 4 | Forehand baseline | 9 | Backhand backcourt |
| 5 | Center baseline | | |

Note: The location of the ball is determined by the right hand of An Se-young.

**Table 2. Names and corresponding numbers of techniques used.**

| Technique Name | Serial Number | Technique Name | Serial Number |
|---|---|---|---|
| High Clear | a | Drop Shot | b |
| Kill | c | Spin Net Shot | d |
| Cross Net | e | Push | f |
| Net Shot | g | Net Block | h |
| Fast Power Drive | i | Net Kill | k |
| Lift | m | Serve | j |

shot locations, the techniques used, and also the winning-point shuttle landing locations are recorded.

**3.2.4 Dataset.** In the machine learning training process, it is common to divide the dataset into three subsets: training set, validation set and test set, where the training set is used for model training, the validation set is used for model tuning and parameter selection, and the test set is used to test the model generalization ability [16]. However, due to the small sample size, segmenting a dataset with an otherwise small amount of data may affect the credibility of the assessment model, as well as being subject to a high degree of chance. Therefore, this study uses a relatively simple cross-validation approach to address the problem to some extent and to make full use of the existing dataset. In the actual modeling process of this study, the number of strikes, the scoring and losing location, as well as the technique used in the last three strikes and the serving location is used as the "input" of the model, and the scoring and losing situations as the "output" of the model, and the data set was randomly divided into training set and test set according to the ratio of 90% and 10%. Typical scoring condition datasets are shown in Table 3.

# 4 Predictive modeling and data analytics

## 4.1 Data preprocessing and rational analysis of dataset

**4.1.1 Characteristic correlation analysis of the "input" parameters of the dataset.** The correlation matrix is a visualization tool for statistical analysis and feature selection. In this study, the correlation matrix was constructed to compare the correlation between the "input" parameters in the dataset, and the results are shown in Fig 2. As can be seen from the figure, firstly, there is a significant negative correlation between the region where the 3rd-from-last strikes and the penultimate shot is located and the shot technique used (-0.67 and -0.71, respectively), which suggests a correlation between these two types of parameters, which may provide overlapping information to some extent. The region where the penultimate shot is located and the shot technique used are still treated as independent parameters here, which is in consideration of the need to subsequently analyze the characteristics of the hitting style. The region and technique of the penultimate shot, compared with the number of total shots and the score, do not have significant strong correlation with each other, and can all be treated as independent

**Table 3. Typical scoring condition dataset.**

| Numbers of Strikes | Location of Winning or Losing | Location of the 3rd last strikes | Technique used in the 3rd last strikes | Location of the penultimate strikes | Technique used in penultimate strikes | Location of the last strikes | Technique used in the last strikes | Scoring Result |
|---|---|---|---|---|---|---|---|---|
| 7 | Area 6 | 3 | g | 8 | c | 1 | k | yes |
| 8 | Compulsory | 7 | b | 7 | c | 2 | g | yes |
| 17 | Area 9 | 3 | m | 9 | b | 7 | i | yes |
| Input | | | | | | | | Output |

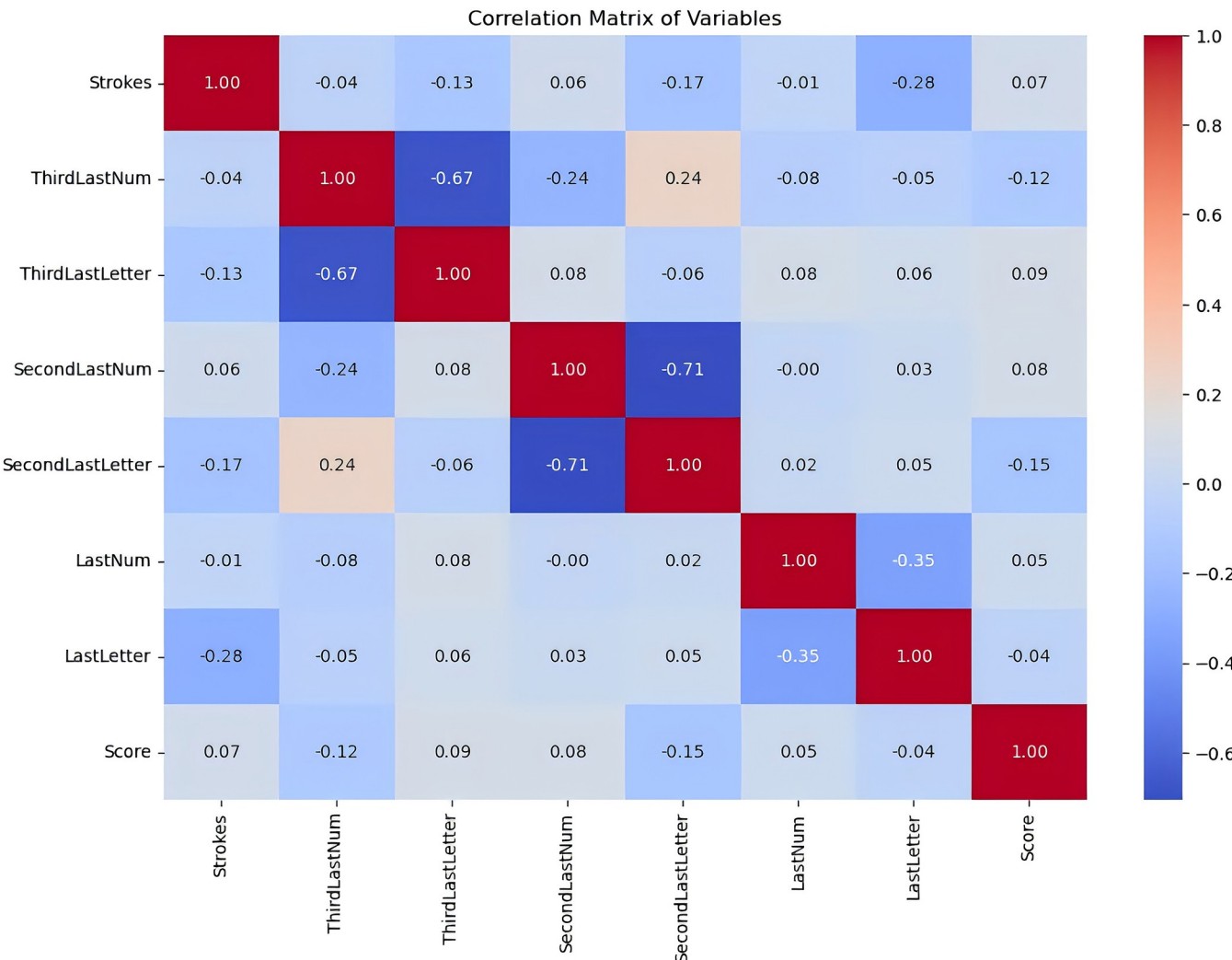

**Fig 2. Correlation matrix between the parameters of the dataset "input".** (The color of each grid in the figure indicates the magnitude of the correlation coefficient between the two variables, with colors closer to red indicating a higher degree of positive correlation, closer to blue indicating a higher degree of negative correlation, and colors closer to white indicating no or little correlation).

predictors, which can have a unique contribution to the prediction of match results. Therefore, using the above parameters as "inputs" to the dataset, there is a certain degree of correlation between them, and at the same time, each of them can provide a unique contribution to the prediction of the results, which is a reasonable way to categorize the dataset.

**4.1.2 Rationalization of data set "outputs".** The principal component analysis (PCA) method can intuitively understand the feature distribution of the data set, and initially determine whether the data set has learning value [16]. Using the principal component analysis method, the "input" and "output" multidimensional data of the dataset are downscaled to a two-dimensional space, and then clustered using K-means, and the results are shown below in Fig 3. It can be seen that the two clusters of gain and loss score are relatively separated in a two-dimensional space, which means that the K-means algorithm can classify the data into two different clusters according to the intrinsic structure of the data, i.e., the dataset contains the intrinsic features that can be learned by the machine learning model, and this method can be used to verify whether the machine learning can successfully learn the features.

## 2D PCA

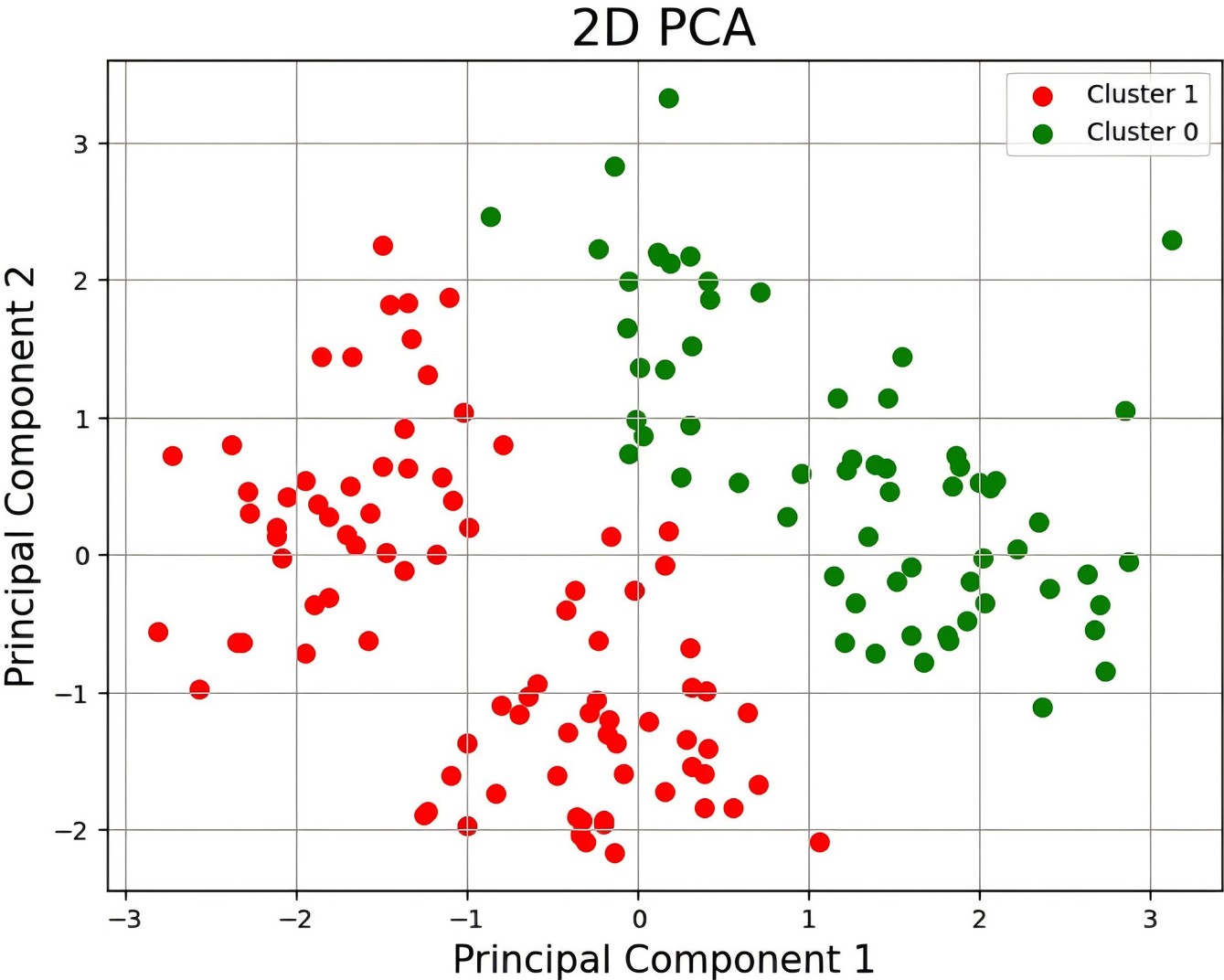

**Fig 3. Results of principal component analysis method.** (Class Cluster 1 and Class Cluster 2 in the figure are separated point distributions after fusing 8 parameters, Principal Component 1 and Principal Component 2 represent the coordinates of the data in the new space, they are the two most important directions in the data, there are better clusters with clearer boundaries, and they are able to capture the maximum data variance).

### 4.2 Machine learning models predict results

Five machine learning algorithms, including decision tree, random forest, XGBoost, support vector machine, and K-nearest neighbor algorithm, were used to train and predict the course of 10 matches between An Se-young and Chen Yufei, Yamaguchi, Dai Ziyin, and Carolina Marin in 2023, and the prediction results are shown in Table 4.

**Table 4. Prediction results of different machine learning methods.**

| Machine Learning Methods | Accuracy |
| --- | --- |
| Decision Tree | 62.50% |
| Random Forest | 62.50% |
| XGBoost | 56.25% |
| Support Vector Machines | 87.50% |
| K-nearest Neighbors | 75.00% |

## SVC Kernel Function Performance Comparison

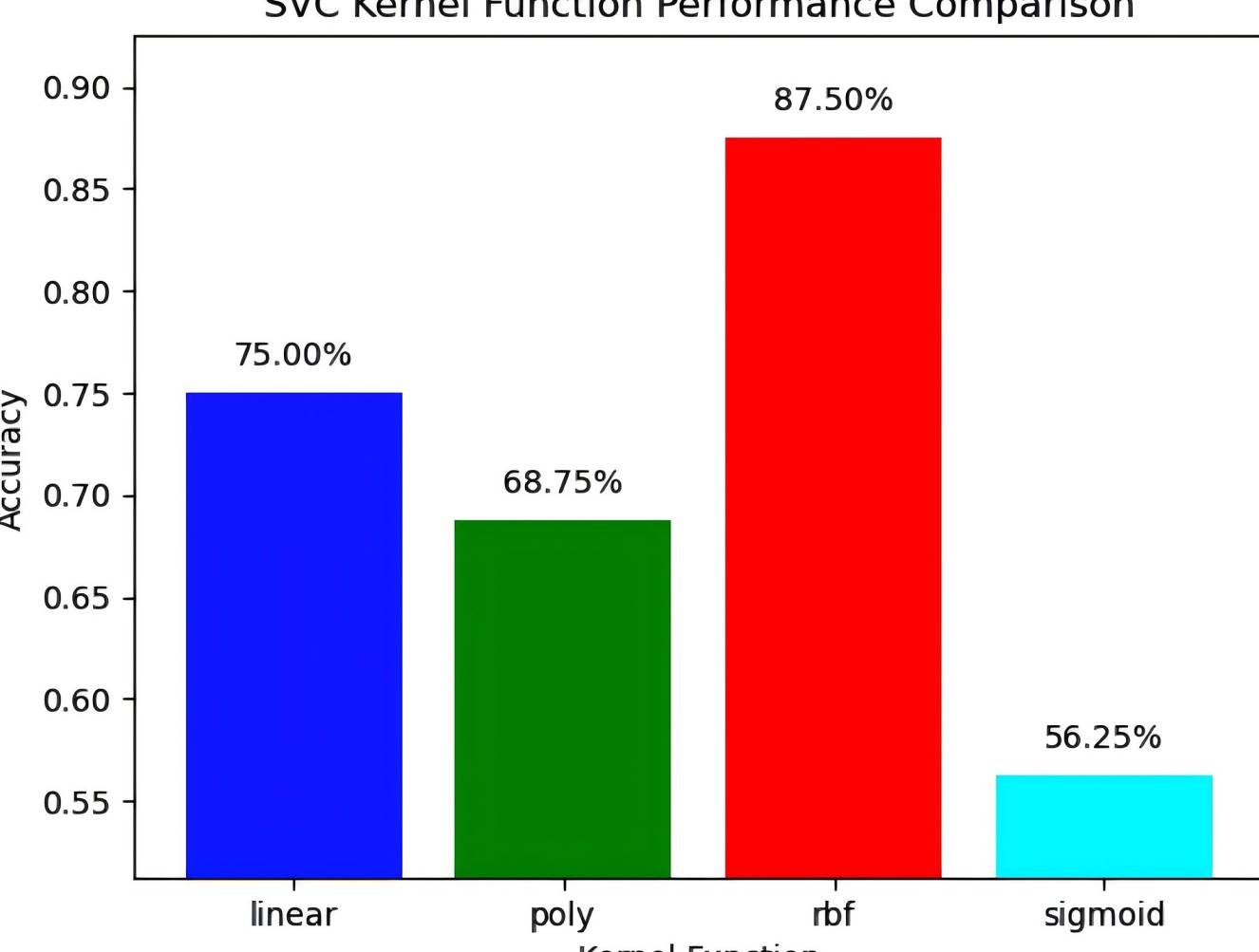

**Fig 4. Prediction results of different kernel functions for support vector machine.**

It can be seen that when the support vector machine uses the RBF function kernel, the accuracy reaches the highest 87.5%, and the consistency of this prediction model is strong (the prediction results of different kernel functions are shown in Fig 4). Among them, the accuracy of the linear kernel is 75.00%, which indicates that the classification method of the dataset described in this paper has obvious regularity and has a strong linear relationship with the scoring/losing of points. The polynomial kernel function and the Sigmoid kernel function are less accurate, which may be due to the fact that the feature relationships of the dataset of the former do not fully conform to the nonlinear pattern assumed by the polynomial kernel, or the parameters of the kernel (such as degrees) are not optimized, and the latter is usually used as an activation function in dichotomous problems rather than the kernel function of the classifier.

The second highest prediction accuracy is the K-nearest neighbor algorithm, with an accuracy of 75.00%. The prediction accuracy of decision tree, random forest and XGBoost method are 56.25%, 62.5% and 62.5%, respectively. The decision tree algorithm indicates that the dataset has a certain complexity, and a single decision tree may not be enough to capture all the

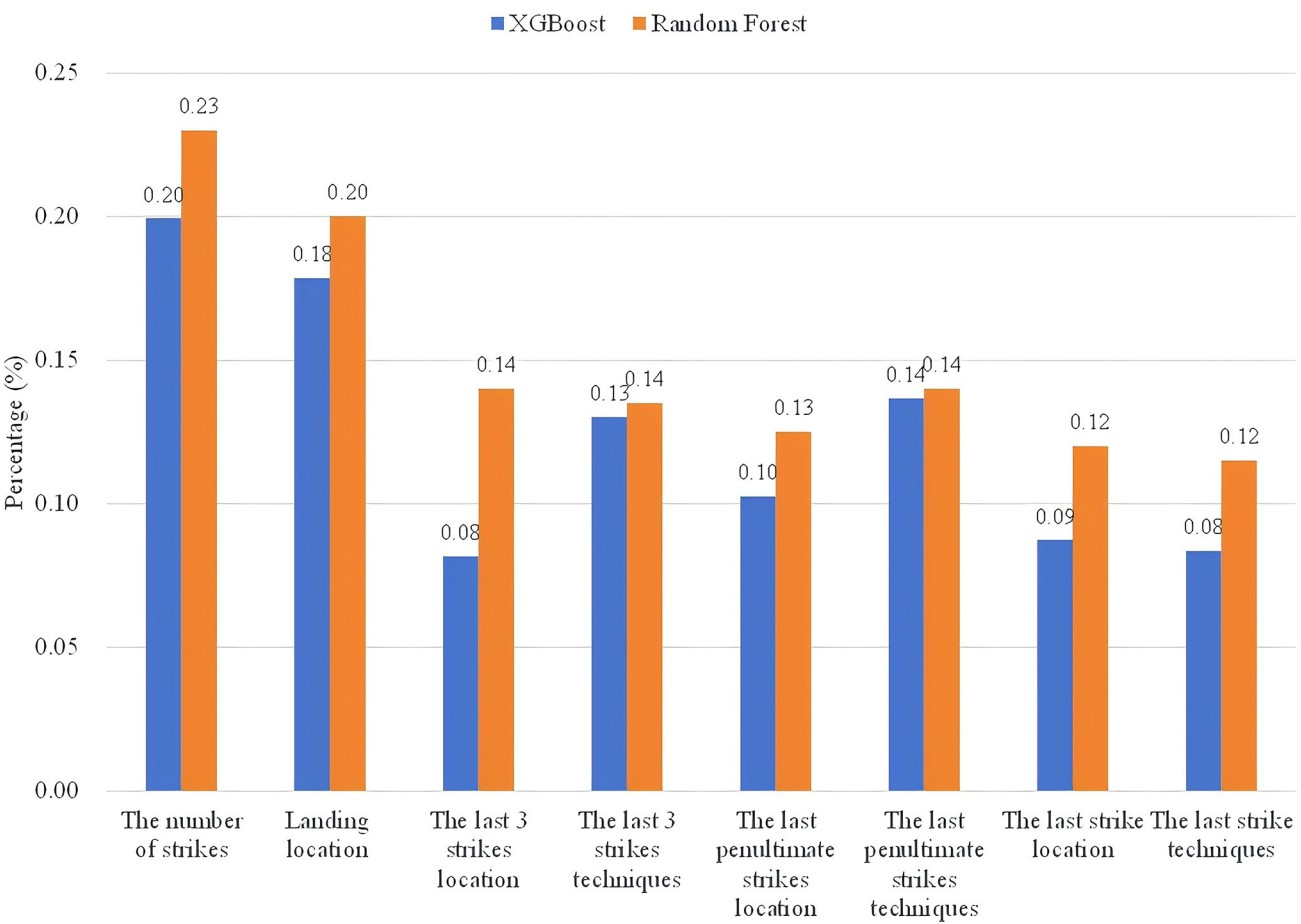

**Fig 5. Random forest and XGBoost predict the importance of "input" parameters.**

patterns, and the poor performance of random forest and XGBoost may be due to the low data dimension, and would be enhanced by a larger data set.

## 4.3 Analysis of the technical and tactical characteristics of An Se-young

Although Random Forest and XGBoost are less accurate than Support Vector Machines and K-Nearest Neighbor algorithms in predicting scores and losses, their ability to characterize the importance of each "input" parameter is useful in analyzing An's technical and tactical play. Therefore, based on the above two models, the relationship between each "input" parameter and "output" in the data set is trained and learned, and the results are shown in Fig 5.

As seen in the above figure, these two methods analyzed the importance of each parameter with the same pattern, except for the region where the penultimate strike is located. The number of strikes in a round is of the highest importance, followed by the landing location where the points are scored or lost. Of the last three shots, the techniques used on the penultimate and second shots are relatively more important. The serving zone is more important when the game comes to the last strike. In the following, the authors have counted the number of strikes for both the scoring round and losing round by An Se-young, the techniques used, and the zone of strikes, and analyzed the characteristics of her playing style in combination with the predictions of Random Forest and XGBoost.

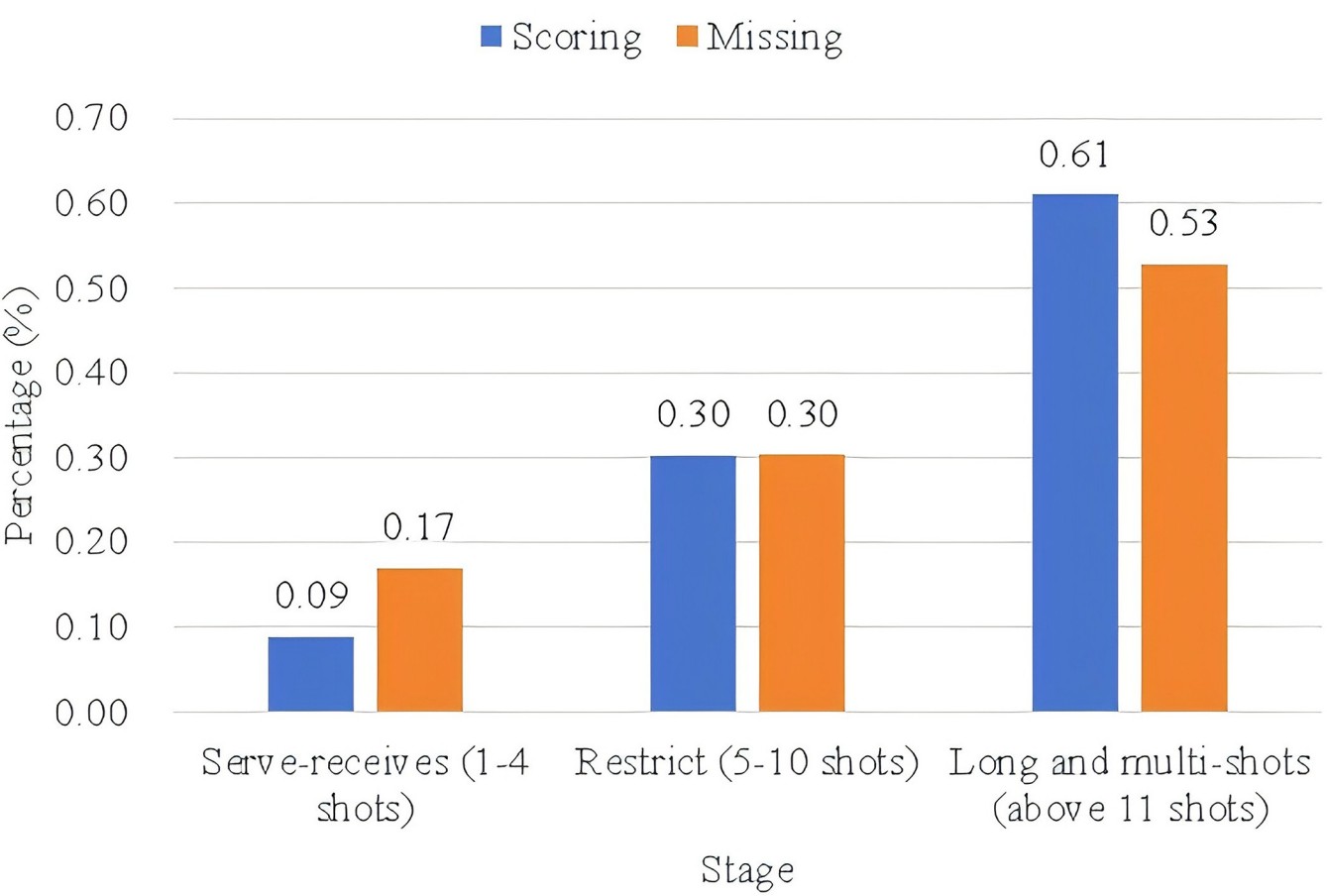

**Fig 6. Statistics on the number of strikes in different rounds by An Se-young.**

**4.3.1 Number of strikes.** The statistics of the shots from different rounds are shown in Fig 6. It can be seen that the distribution of the number of shots is relatively consistent, from the serve to the long multi-shot showing an increasing trend, more than 50% of the rounds in the game are completed by long and multi-shots, and this proportion reaches 61% in the scoring round. The proportion of long multi-taps in conceding rounds is reduced to 53%, while the proportion of opponents scoring points through serve-receives has been increased to 17%. Therefore, judging from the number of shots in the rounds alone, dragging the match into a long and multi-shot battle is a beneficial & notable characteristic of An Se-young's playing style.

**4.3.2 Tactical combinations in the last three strikes.** *(1) Use of techniques.* Figs 7 and 8 count the techniques used by An Se-young in the last three shots when scoring or losing points. In scoring situations, the penultimate shot technique had the highest percentage of keeping the opponent at the baseline when utilizing a high clear or lifting the ball (26%), followed by smashing (14%), then by using a drop shot (11%) and finally by using the net kill (9%). The final shot was still mostly scored by high clear or lifting the ball, and the percentage using this technique also increased from the penultimate shot (35%); followed by drop shot (11%); and then net shot (9%). Starting from the penultimate shot, 60% of An Se-young's organized shots from the backcourt ultimately achieved points through lifting or utilizing a high clear. Secondly, the other common means of scoring such as smashing and net kill appeared mostly in the penultimate shot in An Se-young's style of play, and then decreased in order

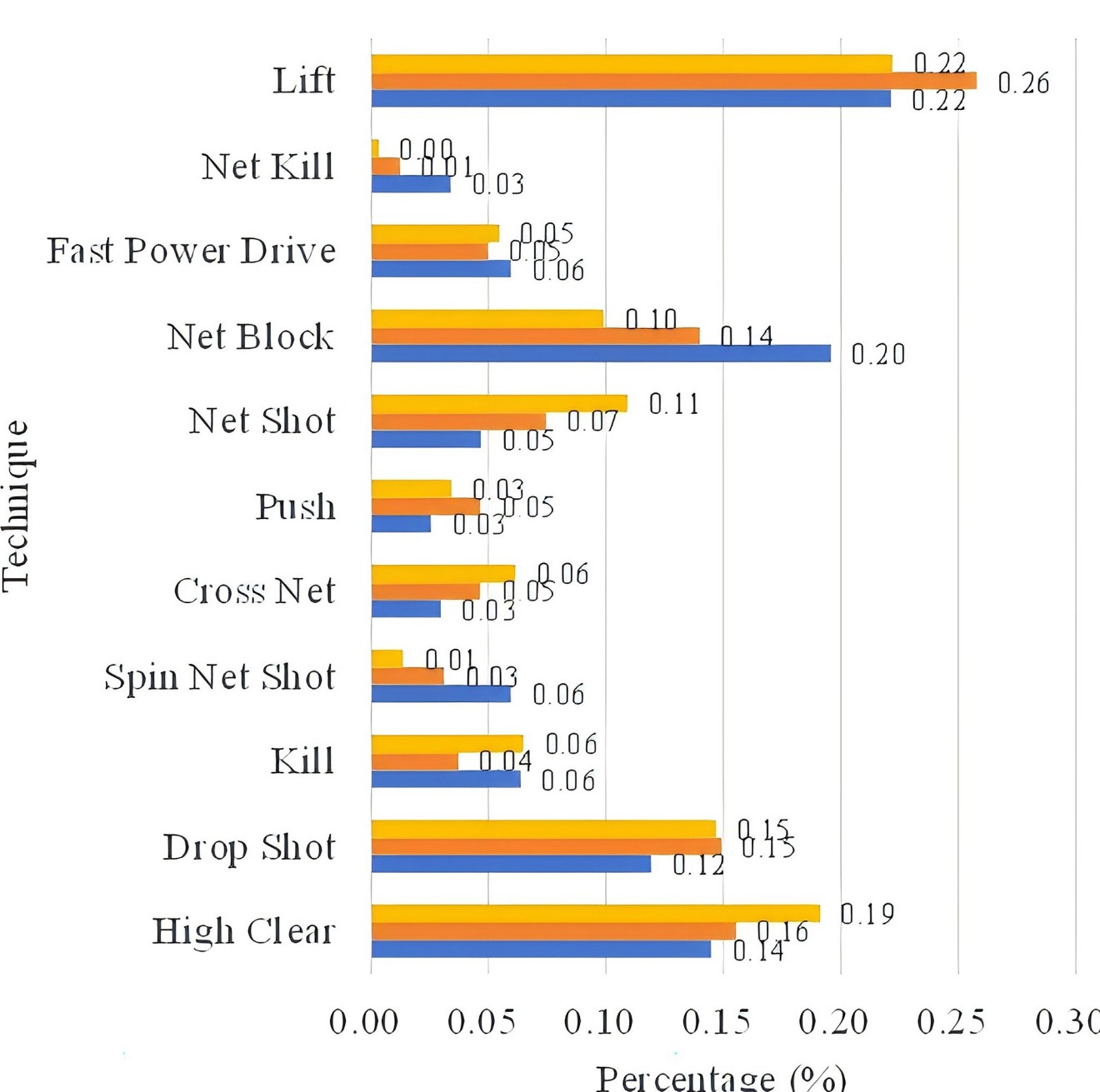

**Fig 7. An So-young's scoring situation the last 3 shots techniques.**

from the penultimate shot to the last shot, indicating that its attacking strength is weaker than the ability to stalemate each other, and it is difficult to score in a single shot, but it is usually possible to create an opportunity through a blitz, and then score during the following 2~3 shots.

The use of techniques for lost points was generally similar to the scoring, with an increase in the proportion of blocks and lobs on the penultimate shot, at 20% and 22% respectively, and

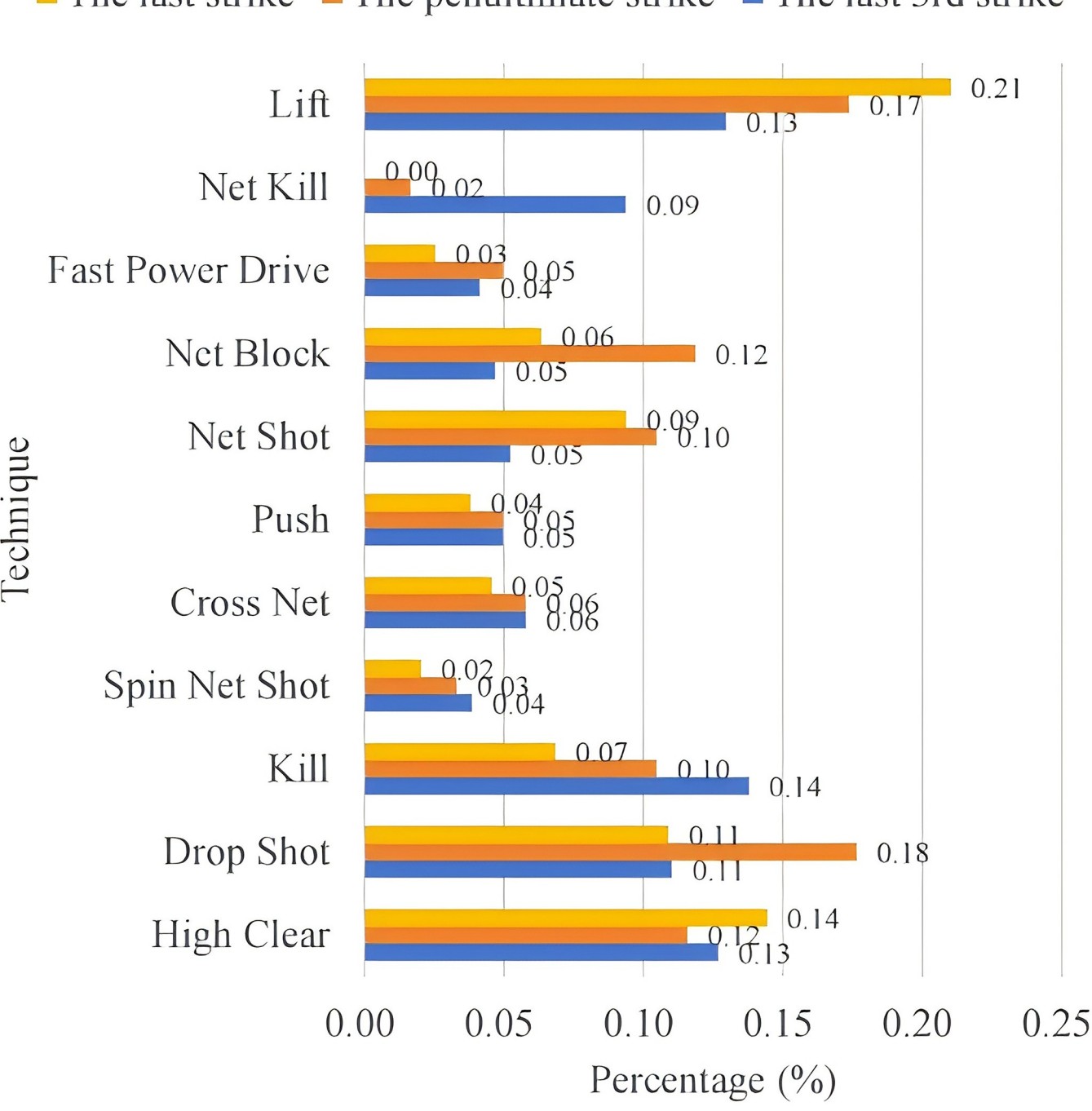

**Fig 8. An So-young's missing situation the last 3 shots techniques.**

a significant decrease in net kill and smash, at 3% and 6% respectively. The proportion of fast drive and net shot releases increased on the last shot, at 5% and 11%. It can be seen that from the penultimate shot onwards, An forced her opponent to lift up the ball by means of blocking the net in order to organize the score tactic from the back court.

*(2) Position of serving.* Figs 9 and 10 count the location of the last three shots of the ball in the case of scoring and missing points by An Se-young. In the scoring situation, the

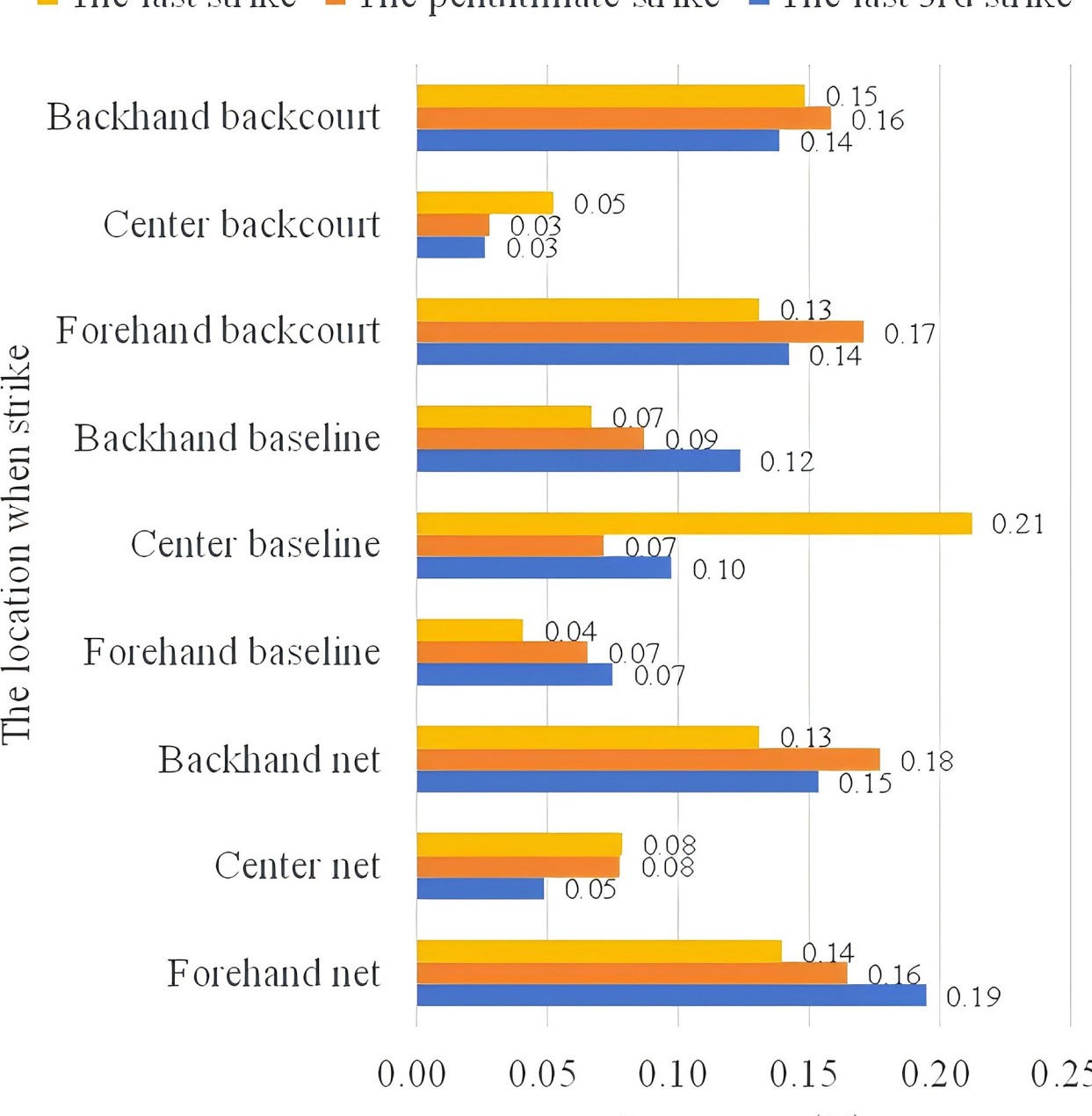

**Fig 9. The last 3 shots area of An So-young's scoring.**

penultimate shot organizes the tactic from both the forehand and backhand to the forecourt and backcourt areas. The probability for An is higher in the forecourt and forehand area (19%) than in the backhand area (15%), while the backcourt has the same percentage at both ends, at 14%. The last shot was scored mainly in the center line center court position with a probability of 21%. On average, the frontcourt and backcourt positions are the main scoring positions, both being 13%~15%.

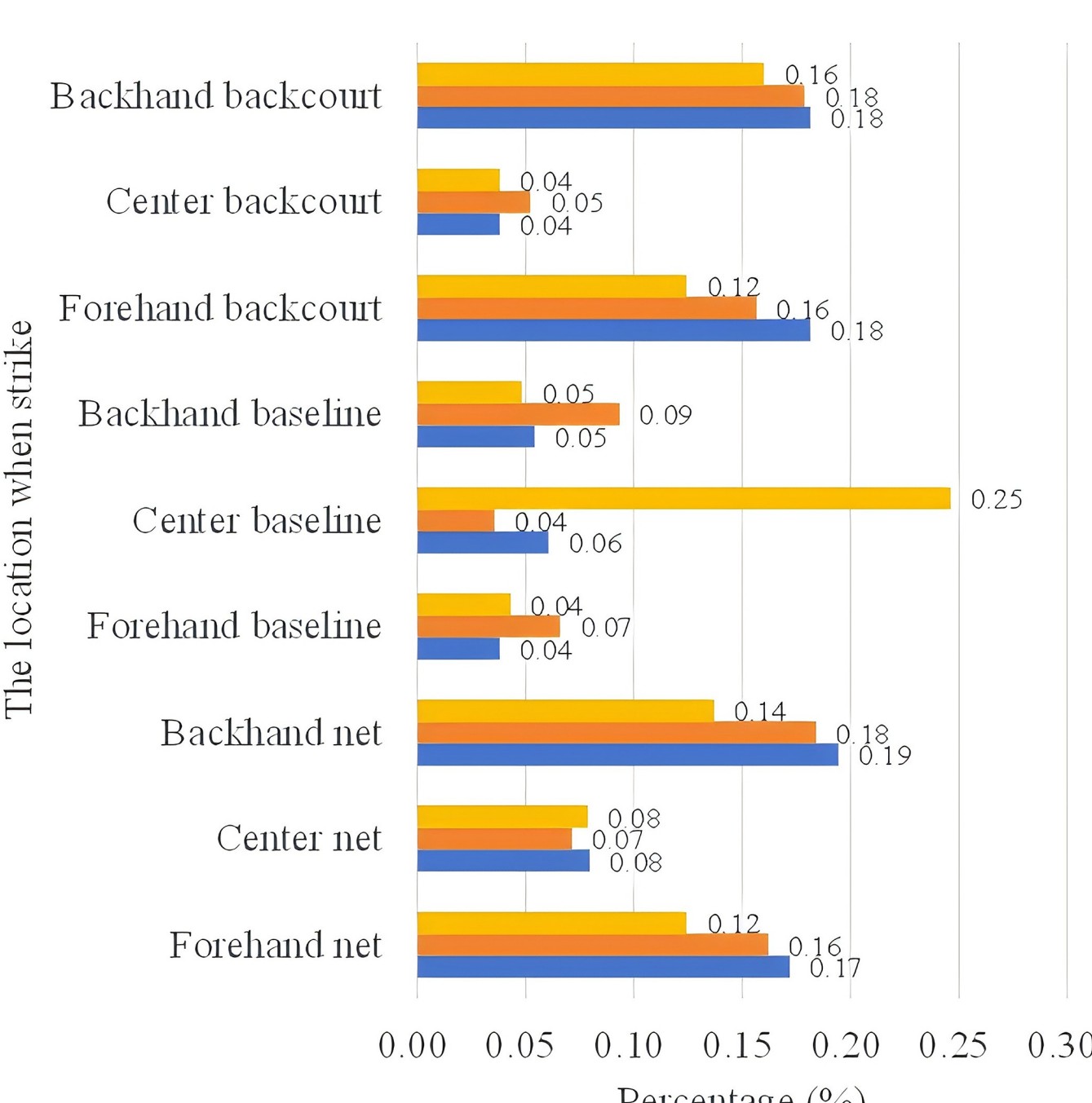

**Fig 10. The last 3 shots area of An So-young's missing.**

The use of striking techniques for points lost was generally similar to points gained, with a significant drop in the percentage of penultimate shots from forehand and backhand midcourt, at 4% and 5% respectively.

Combining the Random Forest and XGBoost algorithm shows the importance of the number of shots and the tactical placement of the last three shots in particular. In the above analyzed data, we can see that, firstly, An Se-young's scoring model is similar to the pattern of her loss model, it also shows how An's playstyle is sustained and unified and has a balanced use of her forehand and backhand in both the front and back half-court areas. Secondly, An Se-young's playing style gradually accumulates advantages over the course of multiple shots, and she creates scoring opportunities through a sudden change of shot speed, which increases her chance of scoring in the following 2~3 shots. The landing point of the final shot is vital in deciding whether An scores or not. Third, regardless of if An Se-young plays the ball in the forecourt or the backcourt, she likes to control the ball on both sides of the opponent's backcourt, and from there begins to organize the ball's path tactically, not seeking to pressure her opponent continuously until the last ball, surprising them.

## 5 Conclusions

This study focuses on An Se-young, the top-ranked female badminton player on the 2023 BWF Olympic Points Table, with a win rate of 89.5%. We successfully developed a machine learning-based prediction model and tactical analysis framework for women's singles badminton. By incorporating the scoring and losing shot placement areas and the tactical details of the last three shots, we improved the traditional "three-stage" data classification method and established a high-quality dataset. Experiments show that this dataset effectively supports machine learning models in learning intrinsic features, enabling accurate predictions of match outcomes.

Among various algorithms compared, the Support Vector Machine (SVM) with an RBF kernel exhibited the best performance, achieving a prediction accuracy of 87.5%, validating the model's effectiveness. Further analysis reveals that An Se-young's tactical style is stable and consistent, primarily focusing on rallying, with balanced use of forehand and backhand, as well as front and backcourt techniques. Her strategy emphasizes gradually establishing an advantage by controlling the backcourt rather than directly scoring through smashes. Ultimately, she creates scoring opportunities through prolonged rallies and precise attacks, particularly emphasizing the placement of the final shot.

In designing the database for this study, we prioritized representativeness while minimizing influencing factors, thus streamlining the sample size. In the future, we plan to expand the data sources and volume, introduce more complex models, like Convolutional Neural Networks (CNNs), and consider dynamic match factors to provide more accurate and efficient guidance for athlete training in sports such as badminton, tennis, and table tennis.

## Supporting information

**S1 Raw data.**
(XLSX)

## Author Contributions

**Data curation:** Yulu Bin.

**Resources:** Kairan Yang.

**Supervision:** Yaodong Wang.

**Writing – original draft:** Hanguang Yuan.

**Writing – review & editing:** Hanguang Yuan.

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
