## [Decision Letter · Decision Letter 0]

1 Jul 2024

PONE-D-24-18327Prediction model and technical and tactical decision analysis of women's badminton singles based on machine learningPLOS ONE

Dear Dr. Wang,

Thank you for submitting your manuscript to PLOS ONE. After careful consideration, we feel that it has merit but does not fully meet PLOS ONE’s publication criteria as it currently stands. Therefore, we invite you to submit a revised version of the manuscript that addresses the points raised during the review process.

We look forward to receiving your revised manuscript.

Kind regards,

Rasool Abedanzadeh, Ph.D

Academic Editor

PLOS ONE

3. In your Methods section, please include additional information about your dataset and ensure that you have included a statement specifying whether the collection and analysis method complied with the terms and conditions for the source of the data.

4. We suggest you thoroughly copyedit your manuscript for language usage, spelling, and grammar. If you do not know anyone who can help you do this, you may wish to consider employing a professional scientific editing service. 

A clean copy of the edited manuscript (uploaded as the new *manuscript* file)”.

5. We note that your Data Availability Statement is currently as follows: [All relevant data are within the manuscript and its Supporting Information files.]

Reviewers' comments:

Reviewer's Responses to Questions

**Comments to the Author**

1. Is the manuscript technically sound, and do the data support the conclusions?

Reviewer #1: Yes

Reviewer #2: Yes

2. Has the statistical analysis been performed appropriately and rigorously? 

Reviewer #1: Yes

Reviewer #2: Yes

3. Have the authors made all data underlying the findings in their manuscript fully available?

Reviewer #1: Yes

Reviewer #2: Yes

4. Is the manuscript presented in an intelligible fashion and written in standard English?

Reviewer #1: Yes

Reviewer #2: Yes

5. Review Comments to the Author

Reviewer #1: Overall, this manuscripts attempts to provide a machine learning model to predict outcomes in women's singles badminton matches by utilizing technical and tactical data of a top player in the sport.

Our review of this manuscript indicates that this study presents the main research findings. The reported results have not been published elsewhere. The experiments, statistics, and other analyses are conducted at an acceptable technical standard and described in sufficient detail. However, there are concerns that need to be addressed.

The conclusions are presented in an appropriate manner and are supported by the data. However, the manuscript is not presented in an intelligible fashion and needs to be rewritten in a clear and concise language.

In the file I reviewed, formulas 1, 4, 7, and 11 are unclear and need to be revised and made visible.

Reviewer #2: First of all, we should praise and thank the implementation of this scientific research.

The article is valuable and scientific.

Some suggestions are provided in the form of comments inside the text.

One of the drawbacks of this article is its small sample size.

Limitations and strengths of this article are not mentioned.

There is no discussion and interpretation about the results

By solving these issues, the mentioned article will be able to be accepted

All the best to you.

6. PLOS authors have the option to publish the peer review history of their article (what does this mean?). If published, this will include your full peer review and any attached files.

Reviewer #1: No

Reviewer #2: No

---

## [Author Response · Author response to Decision Letter 0]

26 Sep 2024

To: PLOS ONE

Dear Editor,

I would like to present my thanks to the editors and reviewers for the valuable comments on our manuscript. The revisions, which were made in accordance with the editors’ and reviewers’ concerns and suggestions, are introduced in detail in the attached file, response to the comments. 

I hope that the reviewers and your journal are satisfied with the improved manuscript, and it will be accepted for publication in Water with your help.

Sincerely yours,

Han-guang YUAN，Master of Athletic Training

Yaodong Wang，Professor of Education

Kairan Yang，Doctor of physical education and training

Yulu Bin, Bachelor of Dance Choreography and Directing

---

## [Editor Report · Decision Letter 1]

15 Oct 2024

Prediction model and technical and tactical decision analysis of women's badminton singles based on machine learning

PONE-D-24-18327R1

Dear Dr. Wang,

We’re pleased to inform you that your manuscript has been judged scientifically suitable for publication and will be formally accepted for publication once it meets all outstanding technical requirements.

Kind regards,

Rasool Abedanzadeh, Ph.D

Academic Editor

PLOS ONE

Additional Editor Comments (optional):

ِDear authors,

By reviewing the revised manuscript file and responding to the reviewers' comments, it was found that the respected authors have carefully answered all the concerns of the referees, and in my opinion, the current form of the manuscript is suitable for publication in the journal.
---

## [Editor Report · Acceptance letter]

4 Nov 2024

PONE-D-24-18327R1 

PLOS ONE

Dear Dr. Wang, 

I'm pleased to inform you that your manuscript has been deemed suitable for publication in PLOS ONE. Congratulations! Your manuscript is now being handed over to our production team.

Kind regards, 

on behalf of

Dr. Rasool Abedanzadeh 

Academic Editor

PLOS ONE